# A Child’s Perception of Their Developmental Difficulties in Relation to Their Adult Assessment. Analysis of the INPP Questionnaire

**DOI:** 10.3390/jpm10040156

**Published:** 2020-10-05

**Authors:** Alina Demiy, Agata Kalemba, Maria Lorent, Anna Pecuch, Ewelina Wolańska, Marlena Telenga, Ewa Z. Gieysztor

**Affiliations:** 1Student Research Group of the Developmental Disorders of Children and Youth, Department of Physiotherapy, Faculty of Health Sciences, Medical University, 50-367 Wroclaw, Poland; alina.demiy@student.umed.wroc.pl (A.D.); maria.lorent@student.umed.wroc.pl (M.L.); anna.pecuch@student.umed.wroc.pl (A.P.); ewelina.wolanska@student.umed.wroc.pl (E.W.); marlena.telenga@student.umed.wroc.pl (M.T.); ewa.gieysztor@umed.wroc.pl (E.Z.G.); 2Division Pediatric Propedeutics and Rare Disorders, Department of Pediatrics, Faculty of Health Sciences, Medical University, 50-367 Wroclaw, Poland; 3Laboratory of Clinical Bases of Physiotherapy, Department of Physiotherapy, Faculty of Health Sciences, Medical University, 50-367 Wroclaw, Poland

**Keywords:** children, adult, difficulties, disorders, coordination, focus

## Abstract

This study involved a comparison of the perception of developmental difficulties in a child by the parents, the teacher, and through the child’s self-assessment. Based on the Institute for Neuro-Psychological Psychology (INPP) questionnaire according to S. Goddard Blythe, three groups were examined: schoolchildren, parents, and teachers. Each of them answered a set of 21 questions and assessed the degree of occurrence of a given difficulty for the child on a scale from 0 to 4. The questions concerned psychomotor problems related to balance, motor coordination and concentration, as well as school skills. In total, 49 questionnaires from children and parents and 46 from teachers were used for the study. The mean answer to each question was calculated within the following groups: child–parent, child–teacher, and parent–teacher. The sum of the children’s answer points was significantly higher than the sum of the parents’ answer points (*p* = 0.037). Children assessed their developmental difficulties more strongly than teachers, but this difference was not statistically significant. The individual difficulties of the children were assessed significantly more seriously or more gently than by the National Scientific Conference “Human health problems—causes, present state, ways for the future” speeches by 44 teacher participants on 5 June 2020. Parents and teachers also assessed the children’s difficulties significantly differently (*p* = 0.044). The biggest difference in answers concerned the question of maintaining attention. The obtained results indicate a significant difference in the perception of difficulties occurring in the same child by the teacher and the parent. The child’s behavior in school and home environments may be different and, depending on the requirements, assessed differently. Children perceive their difficulties much more seriously than adults. Talking and the support of adults can make it easier for a child to overcome developmental difficulties.

## 1. Introduction

The appearance of symptoms such as problems with maintaining balance; coordination problems; difficulty with jointing together elements of running, jumping, throwing, and catching a ball; time–space orientation disorder; deep sensibility or kinesthesia (awareness of the arrangement of the body in space and ability to repeat a set motor pattern) in a schoolchild is a clear sign of developmental difficulties that should be considered by parents or legal guardians. Other indications include issues with reading, writing, and mathematical abilities, such as counting and understanding of instructions [1]. All of the aforementioned symptoms increase the risk of dyslexia and can be the reason for psychomotor and social problems in adult life [2,3]. There are preliminary screening tests that enable early detection of problems connected to learning and behavioral or emotional disorders in schoolchildren. These include, among others, the Institute for Neuro-Psychological Psychology (INPP) questionnaire by Goddard Blythe, which allows a profound examination of children in terms of the presence of psychomotor disorders, which, in turn, can be a sign of neuromotor immaturity [4,5]. The use of a questionnaire allows the selection of children who have trouble at school and children who have motoric problems, which indicate disintegrated primitive reflexes [6]. Research with the use of the aforementioned questionnaire was conducted by Grzywniak. According to the author, a child aged 6 or 7 years old gains the neuropsychological maturity for school learning through a correct development and the integration of the primary reflex within the central nervous system [2,4]. The methods of evaluation of retained reflexes became the research objects of not only Goddard Blythe but also of Masgutova who developed a rehabilitative and therapeutic system, Masgutova neurosensorimotor reflex integration (MNRI), with a view to helping patients with neurological and cognitional disorders [5]. Both authors in their methods acknowledge the importance of the incorrect work of structures responsible for the equilibrium and coordinative abilities of the child (cerebellum and central nervous system) in contrast with neonatal reflex [1].

The use of the INPP questionnaire can determine which children have school and motor problems, indicating disintegrated primary reflexes.

Research concerning the perception of difficulties in children is extremely important for both the parent and teacher perspective, and most importantly the children themselves. An adult becomes a witness of the everchanging influence of the environment, that is the school or home, on the behavior of a schoolchild. The foregoing problem arises because of many factors, e.g., the parental attitude, overprotectiveness of the parents or a liberal upbringing style, peer contact, emotional experiences, teacher competences, and the methods of knowledge transfer. Different attitudes will be observed by a parent in a house where the child feels much more at ease and has a greater sense of security and acceptance and a possibility to release emotions in contrast with teacher observations in the school environment where there are top-down rules and time frames regarding the length of the lessons or breaks. With the use of the screening test and observation, the teacher is able to recognize the children with psychomotor disorders [1]. The early pedagogical diagnosis gives the opportunity to take further educational and, if there is such a need, therapeutic steps [4].

The aim of this study was to compare the perception of a child’s developmental difficulties by the parents, a teacher, and through the child’s self-assessment based on an analysis of the INPP questionnaire.

## 2. Materials and Methods

### 2.1. Examined Group

A total of 68 children took part in the research. For comparison, a number of questionnaires were completed; 49 were filled out by children, 49 by parents (72%), and 44 by teachers (74%). A greater number of questionnaires was taken into consideration for the possibility of comparison depending on the analyzed group. Each pupil was rated thrice—by a parent, a teacher, and through the pupil’s self-assessment.

The first treatment group counted 49 children (21 girls and 28 boys). The average age was 8 years. The youngest pupil was 6 years old, and the oldest was 12 years old (SD = 1.63; MED = 8.0; MOD = 6). All of the participants were elementary students. The second group was formed of parents, and the third of teachers.

### 2.2. Questionnaire

The research was conducted with the use of the INPP screening test by S. Goddard Blythe. It comprises 21 questions for which the answers are given on a 5-grade scale (0–4) where 4 means that the disorder is present to a great extent and 0 means a lack of the disorder [7,8,9].

In the questionnaire, each of the groups had to determine on a scale from 0 to 4 the degree of difficulty with which the child copes in day-to-day life. Among them, concentration problems; problems with sitting still, writing, or reading; easy distraction; and motor problems such as swimming, bike riding, or coordination can be distinguished.

Moreover, every child’s result was summed up and categorized into levels, where the larger the sum of the point, the greater the disorder. The aforementioned scale can be seen in Table 1.

### 2.3. Statistical Methods

Statistical analysis was performed using IBM SPSS Statistics version 25 (IBM Corp., Armonk, NY, USA). Means, standard deviation, and medians were calculated. The Mann–Whitney U test was used to compare the two groups in terms of quantitative/ordinal variables. The level α = 0.05 or α = 0.01 was used for comparisons. The effect size was calculated using eta-squared for the Mann–Whitney U test.

## 3. Results

The results were analyzed in three subgroups: child–parent (Table 2), child–teacher (Table 3), and parent–teacher (Table 4). Table 2, Table 3 and Table 4 show the distribution of the average of particular answers to questions between the groups. Statistically significant differences are highlighted in red. In the child–teacher comparison, 10 of the answers show this feature. Similarly, in the parent–teacher group, the answers vary significantly in 10 cases. In the last child–parent column, there are six differences in grading particular difficulties that are statistically significant

Table 5 shows a comparison between the average sums of results and the sum of levels, and the calculated average score in the subgroups. The number of given answers differs significantly. It is the most noticeable in the parent–teacher subgroup where the averages and the division into levels are substantially apart. The parents often assessed the children’s troubles at the first level. Eight pupils more were classified as that level by the parents than those classified as that level by the teachers. The teachers scored the children’s troubles higher, and the children were classified as the second level more by the teachers than by the parents. The difference in the sum of the points is 6.16 (0.57 for the child–teacher subgroup; 5.32 for child–parent). In this group, there is also the greatest difference between the levels, that being 0.38 (child–teacher 0.08; child–parent 0.26). In the remaining groups, the answers are the same or differ insignificantly in at least two aspects.

### 3.1. Child–Teacher Subgroup

In order to compare the answers in the child–teacher group in detail, 44 questionnaires were analyzed. The results are presented in Figure 1 and Figure 2.

The charts show the layout of the children’s and teachers’ answers. There are clear differences between the perception of the problems that the child struggles with (Figure 1), especially in questions 1, 2, and 3. They touch upon the abilities concerning difficulties with sitting still and keeping attention and the child’s ability to stay focused. 

The teachers marked levels 3 and 4, which are “present to a great degree” and “present at a very high intensity”, more frequently, while the children were more likely to give 0, 1, or 2 points (Figure 2). The opposite was observed for questions 4, 5, 8, 9 10, 17, 18, and 19, where teachers marked 0. The questions concerned motor abilities, coordination, motion sickness, the ability to read the analogue clock, and headaches. All of the aforementioned differences are statistically significant (*p* ≤ 0.02). Differences in answers to question 1 are not statistically significant. Moreover, the comparison of the sum of the points and levels is also not significant (*p* ≥ 0.05).

### 3.2. Parent–Teacher Subgroup

In this group, there were 37 analyzed questionnaires.

In the comparison of the parent–teacher group’s answers, there are differences between the answers to questions 2, 3, 4, and 8, where the teachers marked many more 4s than the parents (Figure 3). Those questions were related to motor coordination. 

For the question concerning reading the time, the teachers marked 0 most of the time, in contrast to parents who marked the 0–3 answers (Figure 4). In questions 11, 13, 14, and 18, the parents marked lower answers (0 and 1) much more frequently than the teachers. These questions concerned the issues of writing, rewriting, and mathematical abilities. Question number 18 touched upon headaches. All of the parameters and the comparison of the sum of points and levels show great statistical importance.

### 3.3. Child–Parent Subgroup

In this group, there were 49 analyzed questionnaires.

In the child–parent group, the answers differ the most for questions 17, 18, 19, and 20 (19 and 20, *p* < 0.05), where the parents more commonly gave answers of 0 (Figure 5). The questions concerned reading the time, maintaining attention, headaches, and fatigue. Parents seldomly marked 4 (present at a very high intensity). Furthermore, there are statistically important differences when it comes to questions 8, 10, 11, and 14. These questions touched upon swimming, motion sickness, reading, and mathematical abilities, respectively.

The children’s answers indicated that they have more difficulties with the aforementioned areas than their parents acknowledge (Figure 6). The comparison between the average sums of the points and levels shows great statistical significance (*p* = 0.04 and *p* = 0.03).

## 4. Discussion

The results of our research show that the answers in the parent–teacher group differed substantially in many aspects. The biggest difference was found for the answers given to the question concerning keeping attention, where the provided scales differ by 2 degrees. The results present a significant difference in the way that parents and teachers perceive a child’s abilities. The difference of the teacher’s and the parent’s assessment where the child’s answer is similar to the teacher’s answer might be a result of a parent’s limited awareness concerning the child, who is in a situation that demands focused attention during the class. However, the similarity between the child’s and the parent’s answers and substantially different teacher’s answers might suggest that the teacher perceives the child to be in a difficult situation in the school environment. Different scores in the answers may also reflect the different expectations of the teacher, the child, and the parent. The difference might moreover suggest that the child behaves differently at school than he/she behaves at home. Significant differences appeared in the answers to the questions concerning the child’s ability to catch a ball and the frequency of headaches. There was an incompatibility between the child’s answer and the parent’s answer concerning this issue. The children more often graded themselves higher in the answer to the question about catching the ball and lower when it comes to the answer concerning headaches. This might indicate the children’s understated self-esteem and concealment of experiencing pain such as headaches, or it might suggest a lack of knowledge about how to qualify the frequency of their condition.

Similar research was previously conducted involving parent–teacher groups. There were also studies conducted with the use of other tests in a form of the questionnaires SRD-6, SPE_R, and the SPE IBE scale. The results of our study, however, showed that the teachers’ observations are far more adequate than the data obtained by the use of the questionnaires mentioned above [1]. However, the INPP questionnaire focuses in particular on the children’s difficulties, and it is proven to be an accurate and precise research tool [6]. Furthermore, there are studies that compare groups of children with groups of adults in general, for instance, children who are hospitalized, the nurses who are taking care of them, and the children’s parents. These studies show that the children, though being under their parents’ custody by law, should have the possibility to be consulted about and take an active part in adjusting the treatment they are undergoing, and they should be allowed to receive the information concerning their medical condition [10,11].

The differences in the perception of children’s problems can lead to misunderstanding and create a stressful situation between parents and teachers. However, it could be reduced by analyzing the results of the research conducted with the use of the INPP questionnaire [4]. This allows us to initially discover the children’s difficulties. With the use of the questionnaire, both the parents and the teachers are able to check which aspects of their children’s life they should focus on more and whether the child is in a need of deeper analyses and further diagnosis. This tool can easily limit the problems that the child deals with and suggest therapeutic steps that will inhibit the progress of the problem in the future. The therapeutic activity and further diagnosis can show the child’s autism disorders, which also are the cause of a child’s problems at school and at home [12]. The observation and the reduction of a risk of further progress of dyslexia, problems reading and writing, and other symptoms of language disorders, at an early stage, can eliminate negative consequences that would appear later on [1]. The difficulties might be caused by the survival of the primitive reflexes [6]. They might indicate that the child suffers from mental problems [13]. They could also be an effect of the collision between home and school environment [14]. The atmosphere of home and the atmosphere of school differ substantially. They consist of different components, such as peers, teachers, parents, siblings who simultaneously influence the child’s behavior [15]. To engage in a dialogue about the differences in perceiving children’s difficulties enables us to make better decisions concerning the form of help that the child should receive.

Moreover, other factors that influence the rate of a child’s difficulties should be taken into consideration too. Some other decisive factors that influence the child’s difficulties include financial situation; parents’ educational background; living conditions; access to knowledge, science, and information; and state of health [16]. All of them should be analyzed in a dialogue between the teacher, the parent, and possibly the child [10,11]. 

The main limitation of this study is that small groups were used for the INPP test. Future work could involve a larger group of children with neurological disorders, thus providing information about the children’s and adults’ perception of the children’s developmental difficulties [1]. Future study could also include a sociological interview to determine the parents’ wealth and education.

The conversation about developmental problems between children and the people who support their development should lead to developing the best tactic on how to act in school and home environments. The cooperation between parents and teachers is necessary in order to achieve maximum results while taking into account the child’s individual needs too. This also enables objective comparison of information provided by both of these sources.

## 5. Conclusions

The presented research concerning parents’ and teachers’ perceptions of children and the children’s self-assessment enabled us to draw the following conclusions: 

1. Teachers notice children’s problems with concentration and distraction during the classes substantially more often than the children themselves.

2. Teachers notice writing and copying problems and issues with math skills more often than parents.

3. Children notice their own physical coordination problems and trouble with concentration more often than parents do.

4. Children are perceived differently by their parents, their teachers, and by themselves. Something that is perceived as troublesome by children is not always perceived as problematic by parents or teachers.

The presented conclusions might provide an important reference both for parents and teachers. The integration and support for both of these communities is the key to success in the proper perception of a child in daily life [17].

## Figures and Tables

**Figure 1 jpm-10-00156-f001:**
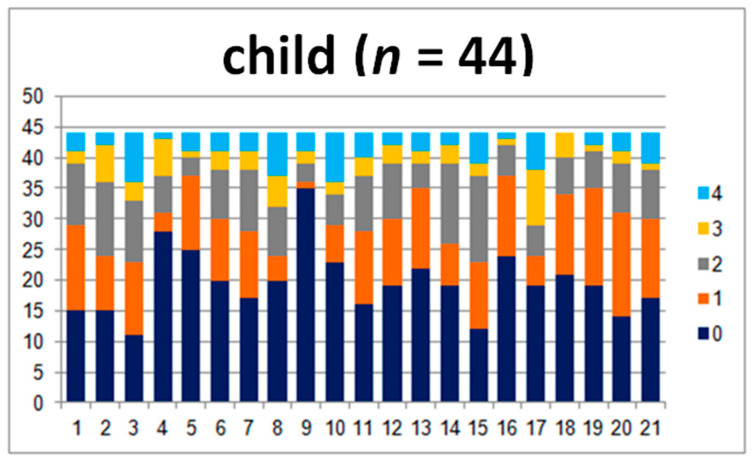
Answers given by the examined children.

**Figure 2 jpm-10-00156-f002:**
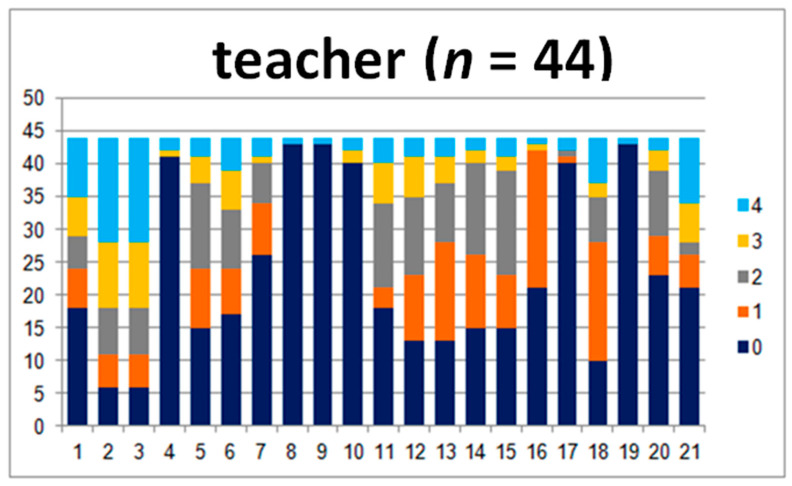
Answers given by the teachers.

**Figure 3 jpm-10-00156-f003:**
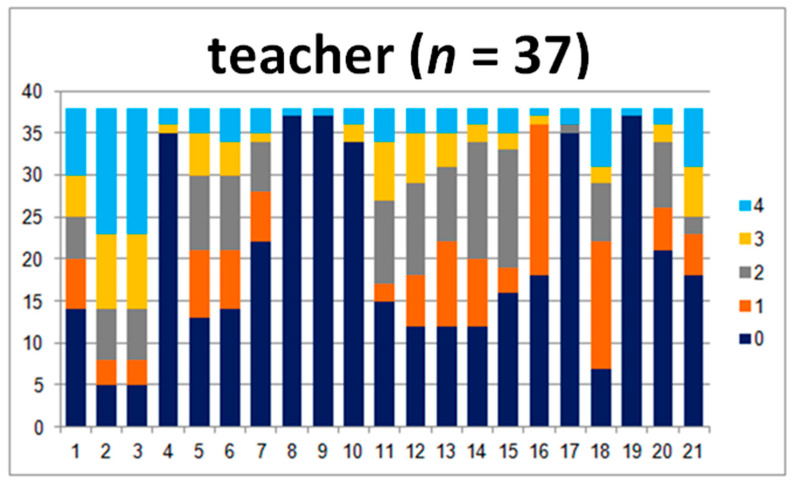
Answers given by the teachers.

**Figure 4 jpm-10-00156-f004:**
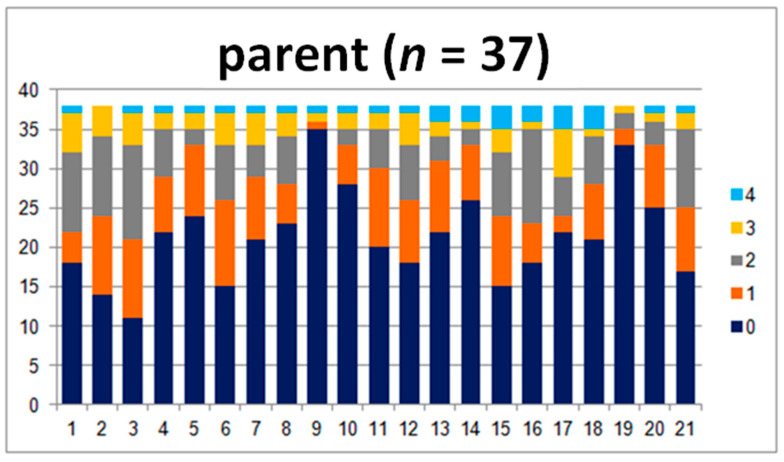
Answers given by parents of examined children.

**Figure 5 jpm-10-00156-f005:**
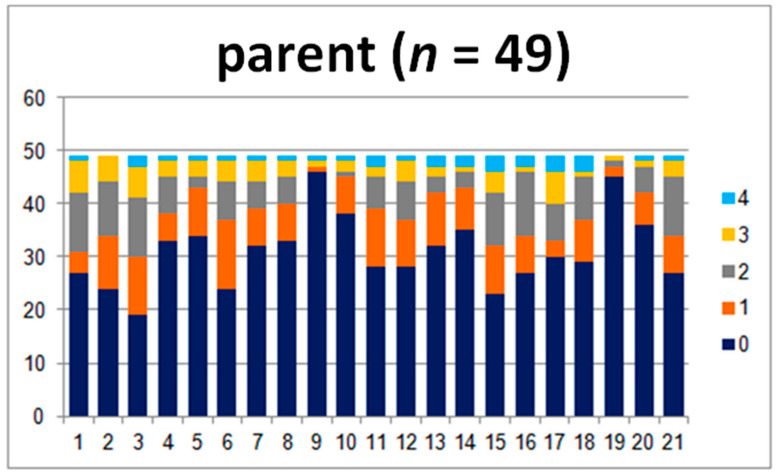
Answers given by parents of examined children.

**Figure 6 jpm-10-00156-f006:**
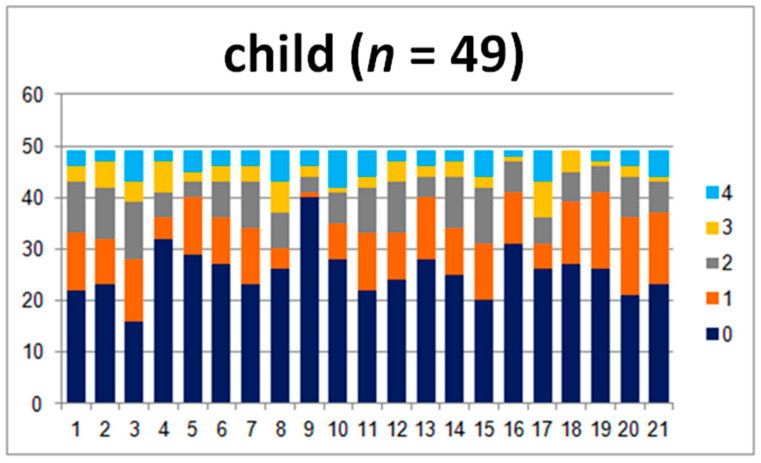
Answers given by examined children.

**Table 1 jpm-10-00156-t001:** The scale of disorder assessment.

Sum	Level	Degree of Disorder
0	0	no disorder
1–21	1	present to a minimum degree
22–42	2	present to a moderate degree
43–63	3	present to a great degree
64–84	4	present at a very high intensity

**Table 2 jpm-10-00156-t002:** The distribution of responses in the child–parent group.

	Children (n = 40)			Parents (n = 40)					
Questionnaire	M	Me	SD	M	Me	SD	U	p	η^2^
1.Inability to sit still	1.2	1	1.2	1.1	1	1.2	786	0.45	0.00
2.Attention problems	1.3	1	1.3	1.1	1	1.1	753	0.33	0.00
3.Easy to distract	1.6	1	1.4	1.4	1	1.1	740	0.28	0.00
4.Coordination problems	1.0	0	1.4	0.8	0	1.1	780	0.35	0.00
5.Incorrect grip	0.9	0	1.3	0.7	0	1.1	744	0.30	0.00
6.Incorrect sitting posture	1.0	1	1.2	1.1	1	1.1	739	0.28	0.00
7.Difficulty catching the ball	1.2	1	1.2	0.8	0	1.2	660	0.09	0.02
8. Difficulty learning to swim	1.4	1	1.5	0.8	0	1.1	642	0.06	0.03
9.Diffiiculty riding a bike	0.5	0	1.2	0.2	0	0.8	701	0.17	0.01
10.Travel sickness	1.1	0	1.5	0.5	0	1.0	638	0.06	0.03
11.Reading problems	1.4	1	1.4	0.8	0	1.1	589	0.02	0.05
12.Writing problems	1.2	1	1.2	0.9	0.5	1.1	697	0.16	0.01
13.Rewriting problems	0.9	0.5	1.2	0.7	0	1.1	718	0.22	0.01
14.Math problems	1.1	1	1.2	0.5	0	1.0	546	0.00	0.08
15.Spelling problems	1.4	1	1.4	1.1	1	1.2	713	0.16	0.01
16.Rearranging numbers or letters	0.7	0	1.0	1.0	1	1.2	692	0.15	0.01
17.Difficulty reading the time	1.5	1	1.6	1.1	0	1.4	669	0.08	0.02
18.Difficulty multi-tasking	0.9	0.55	1.0	0.9	0	1.2	818	0.49	0.00
19.Recurring headaches	1.0	1	1.1	0.2	0	0.7	437	0.00	0.15
20.Frequent fatigue	1.1	1	1.1	0.6	0	1.0	570	0.01	0.06
21.Clear agitation	1.2	1	1.3	1.0	1	1.1	767	0.37	0.00
Sum	23.3	23	13.2	17.3	16	12.1	681	0.05	0.02
Level	1.6	2	0.6	1.4	1	0.7	616	0.04	0.04

* Statistically significant values are marked in red.

**Table 3 jpm-10-00156-t003:** The distribution of responses in the child–teacher group.

	Children (n = 44)			Teachers (n = 44)					
Questionnaire	M	Me	SD	M	Me	SD	U	p	η^2^
1.Inability to sit still	1.18	1	1.17	1.52	1	1.62	909	0.31	0.00
2.Attention problems	1.34	1	1.22	2.57	3	1.44	506	0.00	0.15
3.Easy to distract	1.66	1	1.41	2.57	3	1.44	637	0.00	0.09
4.Coordination problems	0.78	0	1.19	0.25	0	0.94	725	0.02	0.05
5.Incorrect grip	0.75	0	1.14	1.34	1	1.24	680	0.01	0.07
6.Incorrect sitting posture	1.07	1	1.25	1.43	1	1.42	836	0.14	0.01
7.Difficulty catching the ball	1.18	1	1.23	0.80	0	1.19	854	0.05	0.01
8.Difficulty learning to swim	1.43	1	1.55	0.09	0	0.60	471	0.00	0.20
9.Diffiiculty riding a bike	0.57	0	1.23	0.09	0	0.60	795	0.07	0.02
10.Travel sickness	1.23	0	1.57	0.32	0	1.03	610	0.00	0.10
11.Reading problems	1.25	1	1.28	1.43	2	1.39	905	0.30	0.00
12.Writing problems	1.05	1	1.16	1.45	1	1.25	782	0.06	0.03
13.Rewriting problems	0.89	1	1.19	1.30	1	1.19	745	0.03	0.04
14.Math problems	1.12	1	1.19	1.20	1	1.11	945	0.36	0.00
15.Spelling problems	1.46	1	1.28	1.32	1	1.20	945	0.36	0.00
16.Rearranging numbers or letters	0.68	0	0.93	0.64	1	0.81	958	0.47	0.00
17.Difficulty reading the time	1.46	1	1.53	0.25	0	0.89	514	0.00	0.16
18.Difficulty multi-tasking	0.82	1	0.99	1.50	1	1.34	868	0.04	0.01
19.Recurring headaches	0.87	1	1.05	0.09	0	0.60	474	0.00	0.19
20.Frequent fatigue	1.14	1	1.12	0.98	0	1.21	861	0.15	0.01
21.Clear agitation	1.18	1	1.30	1.52	1	1.70	926	0.36	0.00
Sum	23.1	23	12.6	22.70	22	13.30	1003	0.27	0.00
Level	1.61	2	0.65	1.90	2	0.73	794	0.07	0.02

* Statistically significant values are marked in red.

**Table 4 jpm-10-00156-t004:** The distribution of responses in the parent–teacher group.

	Parents (n = 38)			Teachers (n = 38)					
Questionnaire	M	Me	SD	M	Me	SD	U	p	η^2^
1.Inability to sit still	1.13	1	1.23	1.66	1	1.60	591	0.09	0.03
2.Attention problems	1.11	1	1.03	2.68	3	1.42	284	0.00	0.27
3.Easy to distract	1.32	1	1.09	2.68	3	1.42	329	0.00	0.22
4.Coordination problems	0.76	0	1.08	0.29	0	1.01	496	0.01	0.07
5.Incorrect grip	0.61	0	1.00	1.40	1	1.31	462	0.00	0.10
6.Incorrect sitting posture	1.08	1	1.12	1.40	1	1.37	640	0.20	0.01
7.Difficulty catching the ball	0.84	0	1.15	0.87	0	1.26	715	0.47	0.00
8. Difficulty learning to swim	0.79	0	1.14	0.11	0	0.65	463	0.00	0.10
9.Diffiiculty riding a bike	0.21	0	0.81	0.11	0	0.65	685	0.35	0.00
10.Travel sickness	0.51	0	1.02	0.37	0	1.10	602	0.14	0.02
11.Reading problems	0.79	0	1.04	1.55	2	1.45	517	0.02	0.06
12.Writing problems	1.00	1	1.16	1.53	2	1.31	557	0.04	0.04
13.Rewriting problems	0.76	0	1.15	1.37	1	1.26	504	0.01	0.07
14.Math problems	0.58	0	1.08	1.32	1	1.14	431	0.00	0.12
15.Spelling problems	1.21	1	1.28	1.29	1.5	1.29	697	0.40	0.00
16.Rearranging numbers or letters	1.05	1	1.18	0.66	1	0.85	604	0.11	0.02
17.Difficulty reading the time	1.11	0	1.45	0.26	0	0.95	485	0.01	0.08
18.Difficulty multi-tasking	0.90	0	1.25	1.66	1	1.36	462	0.00	0.10
19.Recurring headaches	0.24	0	0.68	0.11	0	0.65	649	0.22	0.01
20.Frequent fatigue	0.55	0	0.95	0.92	0	1.22	613	0.13	0.02
21.Clear agitation	1.00	1	1.09	1.45	1	1.64	648	0.22	0.01
Sum	17.53	16.5	12.17	23.66	22	13.91	503	0.01	0.07
Level	1.29	1	0.65	1.71	2	0.73	489	0.01	0.08

* Statistically significant values are marked in red.

**Table 5 jpm-10-00156-t005:** The average sum of results and levels.

	Child	Teacher	Parent	Teacher	Child	Parent
Average sum of results	23.3	22.73	17.5	23.66	20.16	14.84
Level 0 *	0	0	2	0	6	10
Level 1	22	19	25	17	22	26
Level 2	19	23	9	19	18	11
Level 3	3	0	2	0	3	2
Level 4	0	2	0	2	0	0
Average levels	1.57	1.65	1.32	1.70	1.36	1.1

* Degrees of disorders are described as ”levels”.

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
