# Peer review of "A Child’s Perception of Their Developmental Difficulties in Relation to Their Adult Assessment. Analysis of the INPP Questionnaire"

_jpm, 2020, doi:10.3390/jpm10040156_

Round 1
Reviewer 1 Report
The manuscript by Demiy and colleagues described an assessment comparison by using the INPP questionnaire. The authors analyzed answers of a set of 21 questions of developmental difficulties assessment from a child by oneself, the parent and the teacher. Interestingly, different groups assessed the children’s difficulties differently. The authors analyzed the significant differential answers and concluded the perception variation of the teachers and parents. Overall, the manuscript is concise and clear. It can be further improved by addressing the following questions.
- In Table 2, the authors compared the average of particular answers between teachers, children and parents. The mean is preferred with a central tendency based on the common assumption of a normal distribution of the data. However, Figure 1 to Figure 6 showed score distributions are skewed to one end. Did the authors measure and compare the median value of the data?
- How did the authors calculate statistical significance? Please describe the statistical methods in the methods.
- Did the study have any limitations? How do you plan (future directions) to improve your research against the limitations?
- Different scores in the answers may also reflect a different expectation of the teacher, the child and the parent.
- Minors: please label the question number in Table 2 that is in accord with the number showing in Figure 1 to Figure 6.
Author Response
Please see the attachement.

Reviewer 2 Report
In the abstract, line 26, I am not sure readers will understand the use of the word "seriously". It could refer to how seriously they were in terms of answering the questions or to how serious they thought their developmental issues were. Please clarify.
In Table 1, row for 64-84, there is too much space between "at a very high intensity".
Were any attempts made to assess social desirability response bias? If not, that should be discussed as a limitation.
The last line in the left hand column says "level" but that needs to be more clear even if the table needs a footnote. I think the authors mean "average" but I am not sure.
It would be best if the authors reported effect sizes such as Cohen's d between the pairs of groups so readers could understand the magnitude of the reported effects.
On page 5, the last sentence in the full paragraph is not clear to me in its meaning.
Author Response
Please see the attachement.
